# Sequence element enrichment analysis to determine the genetic basis of bacterial phenotypes

John A. Lees[1,*], Minna Vehkala[2,*], Niko Välimäki[3], Simon R. Harris[1], Claire Chewapreecha[4], Nicholas J. Croucher[5], Pekka Marttinen[6,7], Mark R. Davies[8], Andrew C. Steer[9,10], Steven Y.C. Tong[11], Antti Honkela[12], Julian Parkhill[1], Stephen D. Bentley[1] & Jukka Corander[1,2,13]

Bacterial genomes vary extensively in terms of both gene content and gene sequence. This plasticity hampers the use of traditional SNP-based methods for identifying all genetic associations with phenotypic variation. Here we introduce a computationally scalable and widely applicable statistical method (SEER) for the identification of sequence elements that are significantly enriched in a phenotype of interest. SEER is applicable to tens of thousands of genomes by counting variable-length k-mers using a distributed string-mining algorithm. Robust options are provided for association analysis that also correct for the clonal population structure of bacteria. Using large collections of genomes of the major human pathogens *Streptococcus pneumoniae* and *Streptococcus pyogenes*, SEER identifies relevant previously characterized resistance determinants for several antibiotics and discovers potential novel factors related to the invasiveness of *S. pyogenes*. We thus demonstrate that our method can answer important biologically and medically relevant questions.

[1] Pathogen Genomics, Wellcome Trust Sanger Institute, Cambridge CB10 1SA, UK. [2] Department of Mathematics and Statistics, University of Helsinki, Helsinki FI-00014, Finland. [3] Department of Medical and Clinical Genetics, Genome-Scale Biology Research Program, University of Helsinki, Helsinki FI-00014, Finland. [4] Department of Medicine, University of Cambridge, Cambridge CB2 0SP, UK. [5] Department of Infectious Disease Epidemiology, Imperial College, London W2 1NY, UK. [6] Department of Computer Science, Aalto University, Espoo FI-00076, Finland. [7] Helsinki Institute of Information Technology HIIT, Department of Computer Science, Aalto University, Espoo FI-00076, Finland. [8] Department of Microbiology and Immunology, Peter Doherty Institute for Infection and Immunity, University of Melbourne, Melbourne, Victoria 3010, Australia. [9] Centre for International Child Health, Department of Paediatrics, University of Melbourne, Melbourne, Victoria 3052, Australia. [10] Group A Streptococcal Research Group, Murdoch Children's Research Institute, Parkville, Victoria 3052, Australia. [11] Menzies School of Health Research, Darwin, Northern Territory 0811, Australia. [12] Helsinki Institute for Information Technology HIIT, Department of Computer Science, University of Helsinki, Helsinki FI-00014, Finland. [13] Department of Biostatistics, University of Oslo, 0317 Oslo, Norway. * These authors contributed equally to this work. Correspondence and requests for materials should be addressed to J.C. (email: jukka.corander@helsinki.fi).

The rapidly expanding repositories of genomic data for bacteria hold an enormous and yet largely untapped potential for building a more detailed understanding of the evolutionary responses to changing environmental conditions, such as the widespread use of antibiotics and switches between host-niche as farming practices change.

Studies attempting to determine the genetic basis of bacterial traits have traditionally been limited to identifying emerging clones, which are associated with the phenotype of interest, rather than identifying the specific causal genetic elements[1]. This is partly due to the fact that bacteria reproduce clonally, meaning that a large proportion of the genome is in linkage disequilibrium (LD) with any given trait[2]. The ability of any method to determine which of this large list of variants associated with a trait is truly causal requires that the trait is not uniquely associated with a single clonal lineage. High-recombination rates observed in some species can also break up these large LD blocks, boosting the potential power of an association study to discover the causal variant(s).

For strongly selected traits caused by highly penetrant variants, such as antimicrobial resistance, scanning for homoplasy (convergent evolution) determined by ancestral state reconstruction has been shown to be successful at identifying the causal variant[3]. However, finding variants which are not fully penetrant for a phenotype (as may be the case for clinically relevant traits such as virulence) requires large numbers of samples[4] and a more general test of association.

For these reasons, genome-wide association studies (GWAS) for bacterial phenotypes have only recently started to appear[2,5–8]. Use of standard GWAS methods developed originally for human single-nucleotide polymorphism (SNP) data have been shown to be successfully applicable to core genome mutations in bacteria[6,7]. However, given the high level of genome plasticity of many of the known bacterial species, we can anticipate that such methods can only partially identify genetic determinants of phenotypic variation. To enable discovery of mechanisms related for instance to gene content, alternative alignment-free methods have also been introduced[5,8]. These methods use k-mers, that is, DNA words of length k, as generalized alternatives to SNPs as putative explanations for observed differences in phenotype distributions. The main advantage of k-mers is their ability to capture several different types of variation present across a collection of genomes, including mutations, indels, recombinations, variable promoter architecture and differences in gene content as well as capturing these variations in regions not present in all genomes.

K-mers have been used in bacterial genomics for sequence assembly[9], SNP calling[10] and distance estimation[11]. Previous GWAS studies using k-mers to overcome limitations of SNP-based association have used Monte-Carlo simulations of word gain and loss along an inferred phylogeny to control for population structure[5], whereas SNP-based studies have used clustering algorithms on a core alignment and stratified association tests on the resulting groups of samples[6,7]. The former does not scale computationally to the hundreds of isolates required to find lower effect-size associations, and the latter requires a core alignment, which lacks sensitivity and is difficult to produce when there is a large number of samples, or they are particularly diverse.

Here we present sequence element enrichment analysis (SEER), a method computationally scalable to tens of thousands of genomes, implemented as a stand-alone pipeline that uses either de novo assembled contigs or raw read data as input. We apply SEER to both simulated and the real data from large and diverse populations, and show that it can accurately detect associations with antibiotic resistance caused by both presence of a gene and by SNPs in coding regions, as well as discover novel invasiveness factors.

## Results

**Implementation**. SEER implements and combines three key insights, which we discuss in detail in the methods section: an efficient scan of all possible k-mers with a distributed string mining algorithm, an appropriate alignment-free correction for clonal population structure, and a fast and fully robust association analysis of all counted k-mers.

K-mers allow simultaneous discovery of both short genetic variants and entire genes associated with a phenotype. Longer k-mers provide higher specificity, but less sensitivity than shorter k-mers. Rather than arbitrarily selecting a length before analysis or having to count k-mers at multiple lengths and combine the results, we provide an efficient implementation that allows counting and testing simultaneously at all k-mers at lengths over 9 bases long.

An association test, using an appropriate correction for the clonal population structure, is performed on the counted k-mers. Those reaching significance are filtered post-association and mapped onto both a well-annotated reference sequence and the annotated draft assemblies to allow discovery of variation in accessory genes not present in the reference strain. The significant k-mers themselves can also be assembled into a longer consensus sequence. Annotating variants by predicted function and effect (against a reference sequence) in the resulting k-mers allows fine-mapping of SNPs and small indels.

Meta-analysis of association studies increases sample size, which improves power and reduces false-positive rates[12]. To facilitate meta-analysis of k-mers across studies, the output of SEER includes effect size, direction and standard error, which can be used directly with existing software to meta-analyse all overlapping k-mers.

SEER is implemented in C++, and available at https://github.com/johnlees/seer as source code, a precompiled binary, and a self-contained virtual machine.

**Application to simulated data**. To test the power of SEER across different sample sizes, we simulated 3,069 *Streptococcus pneumoniae* genomes from the phylogeny observed in a Thai refugee camp[13] using parameters estimated from real data including accumulation of SNPs, indels (Supplementary Fig. 1), gene loss and recombination events. Using knowledge of the true alignments, we then artificially associated an accessory gene with a phenotype over a range of odds ratios and evaluated power at different sample sizes (Fig. 1a). The expected pattern for this power calculation is seen, with higher odds ratio effects being easier to detect. Currently detected associations in bacteria have had large effect sizes (OR > 28 host-specificity[5], OR > 3 beta-lactam resistance[6]), and the required sample sizes predicted here are consistent with these discoveries.

The large k-mer diversity, along with the population stratification of gene loss, makes the simulated estimate of the sample size required to reach the stated power clearly conservative. Convergent evolution along multiple branches of a phylogeny for a real population reacting to selection pressures will reduce the required sample size[3].

We also used k-mers counted at constant lengths by DSK[14] to perform the gene presence/absence association (Fig. 1b). Counting all informative k-mers (see Methods) rather than a range of predefined k-mer lengths gives greater power to detect associations, with 80% power being reached at ~1,500 samples, compared with 2,000 samples required by the predefined lengths. The slightly lower power at low sample numbers is due to a stricter Bonferroni adjustment being applied to the larger number of DSM k-mers over the DSK k-mers. This is exactly the expected advantage from including shorter k-mers to increase sensitivity,

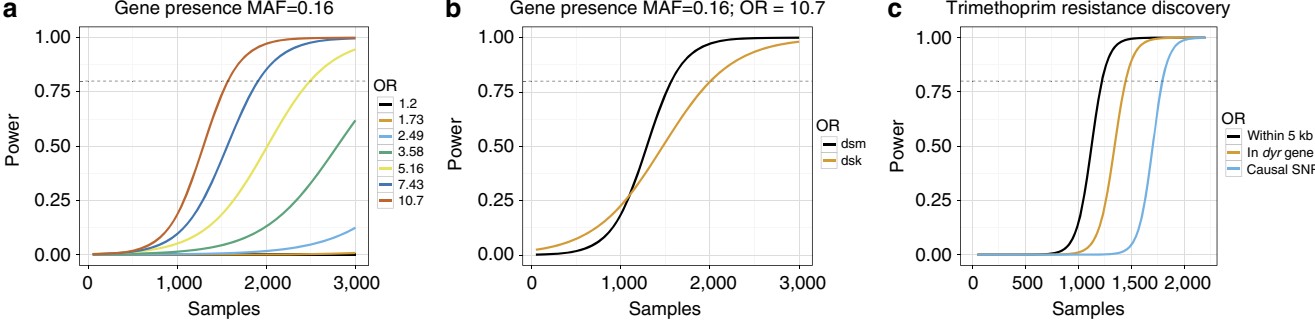

**Figure 1 | Power to find associations versus number of samples.** Using simulations and subsamples of the population as described in the methods, power for (**a**) detecting gene presence/absence at different odds ratios (**b**) using all informative k-mers versus a single length (**c**) detecting k-mers near, in the correct gene, or containing the causal variant for trimethoprim resistance. All curves are logistic fits to the mean power over 100 subsamples.

but as k-mers are correlated with each other due to evolving along the same phylogeny, using the same Bonferroni correction for multiple testing does not decrease specificity.

The strong LD caused by the clonal reproduction of bacterial populations means that non-causal k-mers may also appear to be associated. This is well-documented in human genetics; non-causal variants tag the causal variant increasing discovery power, but make it more difficult to fine-map the true link between genotype and phenotype[15]. In simulations it is difficult to replicate the LD patterns observed in real populations, as recombination maps for specific bacterial lineages are not yet known. To evaluate fine-mapping power of a SNP we instead used the real sequence data and simulated phenotypes based on changing the effect size of a known causal variant and evaluating the physical distance of significant k-mers from the variant site.

Using DSM we counted 68M k-mers which we then tested for association. The 2,639 significant k-mers were mapped to a reference genome, and were found to cover most of the genome with a peak at the causal variant (Supplementary Fig. 2). Mapped k-mers were then placed into three categories: if they contained the causal variant I100L (10 k-mers), were within the same gene (74 k-mers), or within 2.5 kb in either direction (207 k-mers). Figure 1c shows the resulting power when random subsamples of the population are taken. As expected, power is higher when not specifying that the causal variant must be hit, as there are many more k-mers which are in LD with the SNP than directly overlapping it, thus increasing sensitivity.

**Confirmation of known resistance mechanisms in *S. pneumoniae*.** SEER was applied to the sequenced genomes from the study described above[6], using measured resistance to five different antibiotics as the phenotype: chloramphenicol, erythromycin, β-lactams, tetracycline and trimethoprim. Chloramphenicol resistance is conferred by the *cat* gene, and tetracycline resistance is conferred by the *tetM* gene, both carried on the integrative conjugative element (ICE) ICE*Sp*23FST81 in the *S. pneumoniae* ATCC 700669 chromosome[16]. For both of these drug-resistance phenotypes the ICE contains 99% of the significant k-mers, and the causal genes rank highly within the clusters (Table 1, Supplementary Fig. 3).

Resistance to erythromycin is also conferred by presence of a gene, but there are multiple genes that can be causal for this resistance: *ermB* causes resistance by methylating rRNA, whereas *mef/mel* is an efflux pump system[17]. In the population studied, this phenotype was strongly associated with two large lineages (Supplementary Fig. 4), making the task of disentangling association with a lineage versus a specific locus

more difficult. Significant k-mers are found in the mega and omega cassettes, which carry the *mel/mef* and *ermB* resistance elements, respectively.

Hits are also found to other sites within the ICE, a permease directly upstream of *folP*, *prfC* and *gatA*. Macrolide resistance cassettes frequently insert into the ICE in *S. pneumoniae*, so it is in LD with the genes discussed above. In sulphamethoxazole-resistance *folP* is modified by small insertions, with which the adjacent permease is in LD with. Finally, *prfC* and *gatA* are both involved in translation, so could conceivably contain compensatory mutations when *ermB*-mediated resistance is present. Further evidence of these compensatory mutations would be required to rule out the k-mers mapping to them simply being false positives driven by population structure.

Some k-mers do not map to the reference, as they are due to lineage specific associations with genetic elements not found in the reference strain. This highlights both the need to map to a close reference or draft assembly to interpret hits, as well as the importance of functional follow-up to validate potential hits from SEER.

Multiple mechanisms of resistance to β-lactams are possible[6]. Here, we consider just the most important (that is, highest effect size) mutations, which are SNPs in the penicillin binding proteins *pbp2x*, *pbp2b* and *pbp1a*. In this case looking at highest coverage annotations finds these genes, but is not sufficient as so many k-mers are significant—either due to other mechanisms of resistance, physical linkage with causal variants or co-selection for resistance conferring mutations. Instead, selecting the k-mers with the most significant $P$ values gives the top four hit loci as *pbp2b* ($P = 10^{-132}$), *pbp2x* ($P = 10^{-96}$), putative RNA pseudouridylate synthase UniParc B8ZPU5 ($P = 10^{-92}$) and *pbp1a* ($P = 10^{-89}$). The non-*pbp* hit is a homologue of a gene in linkage disequilibrium with *pbp2b*, which would suggest mismapping rather than causation of resistance.

Trimethoprim resistance in *S. pneumoniae* is conferred by the SNP I100L in the *folA/dyr* gene[18]. The *dpr* and *dyr* genes, which are adjacent in the genome, have the highest coverage of significant k-mers (Fig. 2, Supplementary Fig. 2). Following our fine-mapping procedure, we call four high-confidence SNPs that are predicted to be more likely to affect protein function than synonymous SNPs. One is the causal SNP, and the others appear to be hitchhikers in LD with I100L. By evaluating whether sites are conserved across the protein family[19], the known causal SNP is ranked as the highest variant, showing that in this case fine-mapping is possible using the output from SEER.

We then compared the results from SEER with the results from two existing methods (see Methods). The first method (implemented using plink) uses mapping of SNPs against a reference, followed by applying the Cochran–Mantel–Haenszel test at every variable site[6]. The second uses DSK[14] to count

**Table 1 | K-mers associated with antibiotic resistance.**

| Antibiotic | Resistant samples | Number of significant k-mers | | | |
|---|---|---|---|---|---|
| | | Total | Mapped to reference | Highest coverage annotation | Causal element |
| Chloramphenicol | 204 (7%) | 1,526 | 1,526 | 1,508—ICE<br>288—ORF (UniParc B8ZK82)<br>206—rep<br>**166—cat** | 166—*cat* |
| Erythromycin | 803 (26%) | 1,154 | 112 | 10—permease (UniParc B8ZKV5)<br>8—*prfC*<br>6—*gatA*<br>4—ICE | 4—mega element<br>2—*mef*<br>2—omega element |
| β − lactams | 1,563 (51%) | 23,876 | 17,453 | 381—ICE<br>145—prophage MM1<br>50—SPN23F15110 (UniParc B8ZLE7)<br>49—ICE *orf16* | 47—*pbp2x*<br>20—*pbp2b*<br>8—*pbp1a* |
| Tetracycline | 1,958 (64%) | 962 | 962 | 962—ICE<br>136—ICE *orf16*<br>121—ICE *orf15*<br>**96—tetM** | 96—*tetM* |
| Trimethoprim | 2,553 (83%) | 2,639 | 210 | **21—dyr** | 21—*dyr* |

ICE, integrative conjugative element
Results from SEER for antibiotic resistance binary outcome on a population of 3069 *S. pneumoniae*. Significant k-mers are first interpreted by mapping to the ATCC 700669 reference genome. Up to the first four highest covered annotations are shown, and if the known mechanism is amongst these it is highlighted in bold. The ICE is the top hit in three analyses, as it carries multiple drug-resistance elements and is commonly found in multi-drug resistant strains[16]. The distribution of phenotype across the phylogeny is shown in Supplementary Fig. 4.

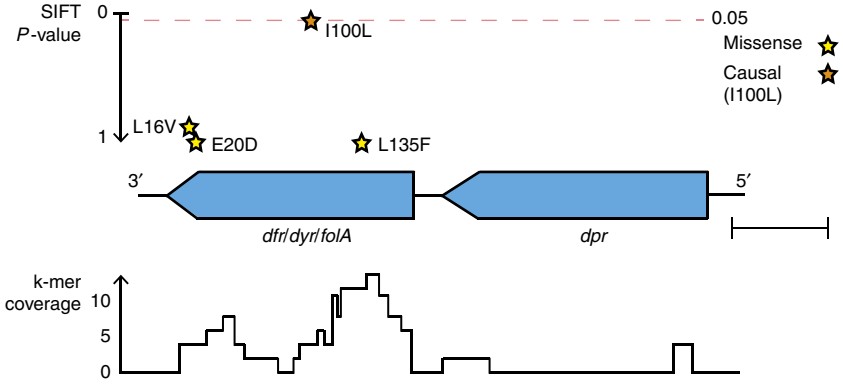

**Figure 2 | Fine mapping trimethoprim resistance.** The locus pictured contains 72 significant k-mers, the most of any gene cluster. Coverage over the locus is pictured at the bottom of the figure. Shown above the genes are high-quality missense SNPs, plotted using their *P* value for affecting protein function as predicted by SIFT. Scale bar is 200 base pairs.

k-mers of length 31, and a highly robust correction for population structure which scales to around 100 genomes[5].

Both SEER and association by core mapping of SNPs (using plink) identified resistances caused by presence of a gene, when it is present in the reference used for mapping (Supplementary Table 1). Both produce their most significant *P* values in the causal element, though SEER appears to have a lower false-positive rate. However, as demonstrated by chloramphenicol resistance, if not enough SNP calls are made in the causal gene this hinders fine-mapping. SNP-mediated resistance showed the same pattern since many other SNPs were ranked above the causal variant. In the case of β-lactam resistance both methods seem to perform equally well, likely due to the higher rate of recombination and the creation of mosaic *pbp* genes.

In addition, as for erythromycin resistance, when an element is not present in the reference it is not detectable in SNP-based association analysis. In such cases, multiple mappings against other reference genomes would have to be made, which is a tedious and computationally costly procedure.

Since the k-mer results from SEER are reference-free, the computational cost of mapping reads to different reference genomes is minimized as only the significant k-mers are mapped to all available references. Alternatively, the significant k-mers can be mapped to all draft assemblies in the study, at least one of which is guaranteed to contain the k-mer, to check whether any annotations are overlapped.

The small sample, combined with fixed length 31-mer approach (see Methods), did not reach significance for chloramphenicol, tetracycline or trimethoprim as the effect size of any k-mer is too small to be detected in the number of samples accessible by the method. Erythromycin had 19,307 hits, and β-lactams 419 hits, at between 1 and 2% minor allele frequency (MAF), which are all false positives that would likely have been excluded by a fully robust population structure correction method.

**Discovery of k-mers associated with *S. pyogenes* invasiveness.** Most bacterial GWAS studies to date have searched for genotypic variants that contribute towards or completely explain antibiotic resistance phenotypes. As a proof of principle that SEER can be used for the discovery stage of sequence elements associated with

other clinically important phenotypes, we applied our tool to 675 *Streptococcus pyogenes* (group A *Streptococcus*) genomes obtained from population diversity studies for genetic signatures of invasive propensity.

We sequenced 347 isolates of *S. pyogenes* collected from Fiji[20] on the Illumina HiSeq platform, and combined this with 328 existing sequences from Kilifi, Kenya[21]. We defined those isolated from blood, cerebrospinal fluid (CSF) or bronchopulmonary aspirate as invasive ($n = 185$), and those isolated from throat, skin or urine as non-invasive ($n = 490$). We ran SEER to determine k-mers significantly associated with invasion, followed by a BLAST of the k-mers with the nr/nt database to determine a suitable reference for mapping purposes. After mapping to this reference SNPs were called (see Methods).

After this preliminary analysis, the top hit was the *tetM* gene from a conjugative transposon (Tn*916*) carried by 23% of isolates (Supplementary Figs 5 and 6). These elements are known to be variably present in the chromosome of *S. pyogenes*[22], and the lack of co-segregation with population structure explains our power to discover the association. However, as a different proportion of the isolates from each collection were invasive (Fiji—13%; Kilifi—43%), the significant k-mers will also include elements specific to the Kilifi data set. Indeed, we found that this version of Tn*916* was never present in genomes collected from Fiji. To correct this geographic bias, we repeat the SEER analysis by including country of origin as a covariate in the regression. This analysis removed *tetM* as being significantly associated with invasiveness, highlighting the importance of such covariate considerations in performing association studies on large bacterial populations.

After applying this correction, we identified two significant hits (Supplementary Fig. 7). The first corresponds to SNPs associating a specific allele of *pepF* (Oligoendopeptidase F; UniProt P54124) with invasive isolates. This could indicate a recombination event, due to the high SNP density and discordance with vertical evolution with respect to the inferred phylogeny[23,24]. The second hit represents SNPs in the intergenic region upstream of both IgG-binding protein H (*sph*) and *nrdI* (ribonucleotide reductase). In support of these findings, previous work in murine models have found differential expression of *sph* during invasive disease[25–27], but little to no expression outside of this niche[28]. If these k-mers were found to affect expression of the IgG-binding protein, this would be a plausible genetic mechanism affecting pathogenesis and invasive propensity[29]. The association of both of these variations would have to be validated either *in vitro* or within a replication cohort, and functional follow-up such as RNA-seq may also aid in elucidating the role of these genetic variants in *S. pyogenes* pathogenesis.

In contrast, application of the existing association methods described above (plink and DSK) to this *S. pyogenes* population data set found no sites significantly associated with invasiveness. The Cochran–Mantel–Haenszel test (stratified by BAPS cluster) that uses SNPs called against a reference sequence failed to identify the *tetM* gene and transposon at these elements are not found in the reference sequence. Furthermore, the population structure of this data set is so diverse that 88 different BAPS clusters were found, which overcorrects for population structure when using the DSK method, leaving too few samples within each group to provide the power to discover associations.

## Discussion

SEER is a reference independent, scalable pipeline capable of finding bacterial sequence elements associated with a range of phenotypes, while controlling for clonal population structure. The sequence elements can be interpreted in terms of protein function

using sequence databases, and we have shown that even single causal variants can be fine-mapped using the SEER output.

Our use of all k-mers 9-100 bases long together with robust regression methods, and the ability to analyse very large sample sizes show improved sensitivity over existing methods. This provides a generic approach capable of analysing the rapidly increasing number of bacterial whole genome sequences linked with a range of different phenotypes. The output can readily be used in a meta-analysis of sequence elements to facilitate the combination of new studies with published data, increasing both discovery power and confirming the significance of results.

As with all association methods, our approach is limited by the amount of recombination and convergent evolution that occurs in the observed population, since the discovery of causal sequence elements is principally constrained by the extent of linkage disequilibrium. However, by introducing improved computational scalability and statistical sensitivity SEER significantly pushes the existing boundaries for answering important biologically and medically relevant questions.

## Methods

**Counting informative k-mers in samples.** We offer three different methods to count k-mers in all samples in a study. For very large studies, or for counting directly from reads rather than assemblies, we provide an implementation of distributed string mining (DSM)[30,31], which limits maximum memory usage per core, but requires a large cluster to run. For data sets up to around 5,000 sample assemblies we have implemented a single core version fsm-lite. For comparison with older data sets, or where resources do not allow the storage of the entire k-mer index in memory, DSK[14] is used to count a single k-mer length in each sample individually, the results of which are then combined.

Over all *N* samples, all k-mers over 9 bases long that occur in more than one sample are counted. All non-informative k-mers are omitted from the output; a k-mer *Z* is not informative if any one base extension to the left (*aZ*) or right (*Za*) has exactly the same frequency support vector as *Z*. The frequency support vector has *N* entries, each being the number of occurrences of k-mer *Z* in each sample. Further filtering conditions are explained in the sections below.

DSM[30,31] parallelizes to as much as one sample per core, and either 16 or 64 master server processes. DSM includes an optional entropy-filtering setting that filters the output k-mers based on both number of samples present and frequency distribution. On our 3,069 simulated genomes this took 2 h 38 min on 16 cores, and used 1 Gb RAM. The distributed approach is applicable up to terabytes of short-read data[31], but requires a cluster environment to run. As an easy-to-use alternative, we propose a single-core version of DSM that is applicable for gigabyte-scale data. We implemented the single core version based on a succinct data structure library[32] to produce the same output as DSM. On 675 *S. pyogenes* genomes this took 3 h 44 min and used 22.3 Gb RAM.

To count single k-mer lengths, an associative array was used to combine the results from DSK in memory. We concatenated results from k-mer lengths of 21, 31 and 41, as in previous studies[5]. This can scale to large genome numbers by instead using external sorting to avoid storing the entire array in memory.

**Filtering k-mers.** Before testing for association we filter k-mers based on their frequency and unadjusted *P* value to reduce false positives from testing under-powered k-mers and reduce computational time.

K-mers are filtered if either they appear in <1% or >99% of samples, or are over 100 bases long. We also test if the *P* value of association in a simple $\chi^2$-test (1 d.f.) is $<10^{-5}$, as in simulations this was true for all true positives, and remove it otherwise. In the case of a continuous phenotype a Welch two-sample *t*-test is used instead.

The effect of this filtering step can be seen by plotting the unadjusted and adjusted *P* values of the k-mers from the simulated data set against each other (Supplementary Figs 8 and 9). Four hundred thirty k-mers of 12.7M passing frequency filtering have an unadjusted *P* value which falls below the $\chi^2$ significance threshold, but would be significant using the adjusted test (and have a positive direction of effect). These k-mers are all short words (10–21 bases; median 12) that appear multiple times per sample, and therefore are of low specificity. Testing the top *P* value k-mer in this set showed a strong association of the presence/absence vector with three population structure covariates used ($P = 1.35e - 24$; $P = 1.15e - 46$; $P = 1.53e - 09$, respectively). Using lasso regression, the first population structure covariate has a higher effect in the model than the k-mer frequency vector (Supplementary Fig. 10). Altogether, this suggests that these filtered k-mers are associated to a lineage related to the phenotype, but are unlikely to be causal for the phenotype themselves. To confirm this, we mapped these k-mers back to the reference sequence. None of these k-mers map to the gene causal to the phenotype.

**Covariates to control for population structure.** To correct for the clonal population structure of bacterial populations, a distance matrix is constructed from a random subsample of these k-mers, on which metric multi-dimensional scaling (MDS) is performed (Supplementary Fig. 11). This is analogous to the standard method used in human genetics of using principal components of the SNP matrix to correct for divergent ancestry[33,34], but has the advantage that no core gene alignment or SNP calling is needed, so can be directly applied to the k-mer counting result. Compared with modelling SNP variation, the use of k-mers as variable sequence elements has been previously shown to accurately estimate bacterial population structure[35].

A random sample of between 0.1% and 1% of k-mers appearing in between 5 and 95% of isolates is taken. We then construct a pairwise distance matrix **D**, with each element being equal to a sum over all $m$ sampled k-mers:

$$d_{ij} = \sum_m \| k_{im} - k_{jm} \|$$

where $k_{im}$ is 1 if the $m$th sampled k-mer is present in sample $i$, and 0 otherwise. Each element $d_{ij}$ is therefore an estimate of the number of non-shared k-mers between a pair of samples $i$ and $j$. Clustering samples using these distances gives the same results as clustering core alignment SNPs using hierBAPS[36] (Supplementary Fig. 12), which has been used in previous bacterial GWAS studies to correct for population structure.

Metric MDS is applied to **D**, projecting these distances into a reduced number of dimensions. The normalized eigenvectors of each dimension are used as covariates in the regression model. The number of dimensions used is a user-adjustable parameter, and can be evaluated by the goodness-of-fit and the magnitude of the eigenvalues. In species tree with two lineages and 96 isolates, one dimension was sufficient as a population control (Supplementary Fig. 13), whereas for the larger collection of 3,069 isolates 10–15 dimensions were needed to give tight control (Supplementary Fig. 14). Over all our studies, generally three dimensions appeared a good trade-off between sensitivity and specificity.

**Logistic and linear regression.** For each k-mer, a logistic curve is fitted to binary phenotype data, and a linear model to continuous data, using a time efficient optimization routine to allow testing of all k-mers. Bacteria can be subject to extremely strong selection pressures, producing common variants with very large effect sizes, such as antibiotics inducing resistance-conferring variants. This can make the data perfectly separable, and consequently the maximum likelihood estimate ceases to exist for the logistic model. Firth regression[37] has been used to obtain results in these cases.

For samples with binary outcome vector $y$, for each k-mer a logistic model is fitted:

$$\log\left(\frac{y}{I - y}\right) = \mathbf{X}\boldsymbol{\beta}$$

where absence and presence for each k-mer are coded as 0 and 1, respectively, in column 2 of the design matrix **X** (column 1 is a vector of ones, giving an intercept term). Subsequent columns $j$ of **X** contain the eigenvectors of the MDS projection, user-supplied categorical covariates (dummy encoded), and quantitative covariates (normalized). The Broyden–Fletcher–Goldfarb–Shanno algorithm is used to maximize the log likelihood $L$ in terms of the gradient vector $\boldsymbol{\beta}$ (using an analytic expression for d(log $L$)/d$\boldsymbol{\beta}$):

$$\log L \propto \sum_i y_i \cdot \log(\text{sig}(\mathbf{X}\boldsymbol{\beta})_i) + (1 - y_i) \cdot \log(\text{sig}(1 - \mathbf{X}\boldsymbol{\beta})_i)$$

where sig is the sigmoid function. If this fails to converge, $n$ Newton–Raphson iterations are applied to $\boldsymbol{\beta}$:

$$\boldsymbol{\beta}_{n+1} = \boldsymbol{\beta}_n + [-L''(\boldsymbol{\beta}_n)]^{-1} \cdot L'(\boldsymbol{\beta}_n)$$

from a starting point using the mean phenotype as the intercept, and the root-mean squared beta from a test of k-mers passing filtering

$$\beta_{0,0} = \frac{\Sigma y_i}{n}$$

$$\beta_{0, j>0} = 0.1$$

which is slower, but has a higher success rate. If this fails to converge due to the observed points being separable, or the s.e. of the slope is $>3$ (which empirically indicated almost separable data, with no counts in one element of the contingency table), Firth logistic regression is then applied. This adds an adjustment to log $L$:

$$\log L(\boldsymbol{\beta})^* = \log L(\boldsymbol{\beta}) + \frac{1}{2} \cdot \log\left|\frac{d^2 L}{d\boldsymbol{\beta}^2}(\boldsymbol{\beta})\right|$$

using which Newton–Raphson iterations are applied as above.

In the case of a continuous phenotype a linear model is fitted:

$$Y = \mathbf{X}\boldsymbol{\beta}$$

The squared distance U($\boldsymbol{\beta}$)

$$U(\boldsymbol{\beta}) = \| y - \mathbf{X}\boldsymbol{\beta} \|^2$$

is minimized using the Broyden–Fletcher–Goldfarb–Shanno algorithm. If this fails to converge then the analytic solution is obtained by orthogonal decomposition:

$$\mathbf{X} = \mathbf{QR}$$

then back solving for $\boldsymbol{\beta}$ in:

$$\mathbf{R}\boldsymbol{\beta} = \mathbf{Q}^T \mathbf{y}$$

In both cases the s.e. on $\beta_1$ is calculated by inverting the Fisher information matrix $d^2 L/d\boldsymbol{\beta}^2$ (inversions are performed by Cholesky decomposition, or if this fails due to the matrix being almost singular the Moore–Penrose pseudoinverse is taken) to obtain the variance-covariance matrix. The Wald statistic is calculated with the null hypothesis of no association ($\beta_1 = 0$):

$$W = \frac{\beta_1}{SE(\beta_1)}$$

which is the test statistic of a $\chi^2$ distribution with 1 d.f. This is equivalent to the positive tail of a standard normal distribution, the integral of which gives the $P$ value.

**Significance cutoff.** For the basal cutoff for significance we use $P < 0.05$, which in our testing we conservatively Bonferroni corrected to the threshold $1 \times 10^{-8}$ based on every position in the *S. pneumoniae* genome having three possible mutations[38], and all this variation being uncorrelated. This is a strict cutoff level that prevents a large number of false positives due to the extensive amount of k-mers being tested, but does not over-penalize by correcting directly on the basis of the number of k-mers counted. To calculate an empirical significance testing cutoff for the $P$ value under multiple correlated tests, we observed the distribution of $P$ values from 100 random permutations of phenotype. For the 3,069 Thai genomes setting the family-wise error rate at 0.05 gave a cutoff of $1.4 \times 10^{-8}$, supporting the above reasoning.

In general, the number of k-mers and the correlations between their frequency vectors will vary depending on the species and specific samples in the study, so the $P$ value cutoff should be chosen in this manner (either by considering possible variation given the genome length, or by permutation testing) for each individual study. Association effect size and $P$ value of the MDS components are also included in the output, to compare lineage and variant effects on the phenotype variation.

**SEER implementation.** SEER is implemented in C++ using the armadillo linear algebra library[39], and dlib optimization library[40]. On a simulation of 3,069 diverse 0.4 Mb genomes, 143M k-mers were counted by DSM and 25M 31-mers by DSK. On the largest DSM set, using 16 cores and subsampling 0.3M k-mers (0.2% of the total), calculating population covariates took 6 h 42 min and 8.33 GB RAM. This step is $O(N^2 M)$ where $N$ is number of samples and $M$ is number of k-mers, but can be parallelized across up to $N^2$ cores.

Processing all 143M informative k-mers as described took 69 min 44 s and 23 MB RAM on 16 cores. This step is $O(M)$ and can be parallelized across up to $M$ cores.

On the real data set of full-length genomes the 68M informative k-mers counted was less than the simulated data set above, as the parameters of the simulation created particularly diverse final genomes (Supplementary methods).

**Interpreting significant k-mers.** K-mers reaching the threshold for significance are then post-association filtered requiring $\beta_1 > 0$ as a negative effect size does not make biological sense. Remaining k-mers are searched for by exact match in their *de novo* assemblies, and annotations of features examined for overlap of function. BLAT[41] is also used with a step size of 2 and minimum match size of 15 to find inexact but close matches to a well-annotated reference sequence.

To better search for gene clusters associated with phenotype, these k-mers are assembled using Velvet[9] choosing a smaller sub-k-mer size, which maximizes longest contig length of the final assembly. K-mers that are then substrings of others significant k-mers are removed.

Small k-mers are more likely than full reads to map equally well to multiple places in the reference genome, so reporting both mappings increases the sensitivity. For this data set an average of 21% of k-mers significantly associated with antibiotic resistance report secondary mappings. These k-mers are short (median 15 bp), and therefore have low specificity and high sensitivity as expected.

**Mapping of a single SNP.** Using the BLAT mapping of significant k-mers to a reference sequence, SNPs are called using bcftools[42]. Quality scores for a read are set to be identical, and are set as the Phred-scaled Holm-adjusted $P$ values from association. High-quality (QUAL$>100$) SNPs are then annotated for function using SnpEff[43], and the effect of missense SNPs on protein function is ranked using SIFT[19].

**Comparison with existing methods.** We compare with two existing methods. The first uses a core-genome SNP mapping along with population clusters defined from the same alignment to perform a Cochran–Mantel–Haenszel test at every called variant site[6]. The second uses a fixed k-mer length of 31 as counted by DSK[14], with a Monte Carlo phylogeny-based population control[5]. As the second method is not

scalable to this population size we used our population control as calculated from all genomes in the population, and a subsample of 100 samples to calculate association statistics, which is roughly the number computationally accessible by this method. In both cases, the same Bonferroni correction is used as for SEER.

**Simulating bacterial populations.** A random subset of 450 genes from the *Streptococcus pneumoniae* ATCC 700669 (ref. 16) strain were used as the starting genome for Artifical Life Framework (ALF)[44]. ALF simulated 3,069 final genomes along the phylogeny observed in a Thai refugee camp[13]. An alignment between *S. pneumoniae* strains R6, 19F and *Streptococcus mitis* B6 using Progressive Cactus was used to estimate rates in the GTR matrix and the size distribution of insertions and deletions (INDELs—Supplementary fig. 3). Previous estimates for the relative rate of SNPs to INDELs[45] and the rate of horizontal gene transfer and loss[13] were used.

pIRS[46] was used to simulate error-prone reads from genomes at the tips of the tree, which were then assembled by Velvet[9]. DSM was used to count k-mers from these *de novo* assemblies.

To test the similarity of the population control to existing methods, 96 full *S. pneumoniae* ATCC 700669 genomes were evolved with ALF. Intergenic regions were also evolved using Dawg[47] at a previously determined rate[48]. These were combined, and assemblies generated and k-mers counted as above. A distance matrix was created from 1% of the k-mers as described above, and a neighbour-joining tree produced from this.

The resulting tree was ranked against the true tree by counting one for each pair of isolates in each BAPS cluster, which had an isolate not in the same BAPS cluster as a descendent of their MRCA.

**Simulating phenotype based on genotype and odds ratio.** Ratio of cases to controls in the population ($S_R$) was set at 50% to represent antibiotic resistance, and a single variant (gene presence/absence or a SNP) was designated as causal. MAF in the population is set from the simulation, and odds ratio (OR) can be varied. The number of cases $D_E$ is then the solution to a quadratic equation[49], which is related to probability of a sample being a case by:

$$P_{\text{case}\,|\,\text{exposed}} = \frac{D_E}{\text{MAF}}$$

$$P_{\text{case}\,|\,\text{notexposed}} = \frac{\frac{S_R}{S_R + 1} - D_E}{1 - \text{MAF}}$$

The population was then randomly subsampled 100 times, with case and control status assigned for each run using these formulae. Power was defined by the proportion of runs that had at least one k-mer in the gene significantly associated with the phenotype.

**Code availability.** SEER is available at https://github.com/johnlees/seer, DSM at https://github.com/HIITMetagenomics/dsm-framework and fsm-lite at https://github.com/nvalimak/fsm-lite. Scripts used to perform the simulations are available at https://github.com/johnlees/bioinformatics

**Data availability.** *S. pyogenes* sequence reads are available on the European Nucleotide Archive under study accession IDs PRJEB2839 (isolates from Fiji) and PRJEB3313 (isolates from Kilifi). Results from the *S. pyogenes* invasiveness GWAS can be found at: http://dx.doi.org/10.6084/m9.figshare.1613851 and can be loaded directly into Phandango (http://jameshadfield.github.io/phandango/) to view the results.

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

## Acknowledgements

We thank James Hadfield for his help in integrating SEER's output into the bacterial genome visualization tool Phandango, and Jeff Barrett and his group for helpful discussions on the relation of association studies in human genetics to prokaryotic genetics. This work was supported by Wellcome Trust grants 098051 and 107376/Z/15/Z, MRC grant 1365620, ERC grant 239784, Academy of Finland grant 287665 and COIN Centre of Excellence.

## Author contributions

J.A.L.—designed methods, performed analysis and wrote manuscript. M.V.—designed methods, performed analysis and wrote manuscript. N.V.—Participated in method design, edited manuscript. S.R.H.—interpretation and preparation of *S. pyogenes* data. C.C.—prepared genetic and metadata from Maela isolates. N.J.C.—helped with interpretation of antibiotic resistance elements, edited manuscript. P.M.—participated in method design and edited manuscript. A.H.—participated in method design, edited manuscript. M.R.D.—analysis of *S. pyogenes* data and edited the manuscript. A.C.S.—collection of *S. pyogenes* isolates from Fiji, edited the manuscript. S.Y.C.T.—culturing and extraction of *S. pyogenes* isolates from Fiji and edited the manuscript. J.P.—advised on microbiological interpretation, edited the manuscript. S.D.B.—advised on microbiological interpretation and edited the manuscript. J.C.—designed method, performed analysis and wrote the manuscript.
