## [Peer Review File · Nature Communications]

Reviewers' comments:

Reviewer #1 (Remarks to the Author):

This paper appears to offer several improvements on the general kmer approach to GWAS with large samples of bacterial genomes. Simulations are used to characterize the method, and two streptococcal samples are used to demonstrate its use with real data. My major concerns should be addressable with moderate effort. Overall, I feel the approach could be a useful addition to the bacterial GWAS toolbox.

Major concerns:

1) The authors did not clearly communicate some of the details related to the population structure correction, pre-processing of kmers, and P-value threshold for significance. During the presentation in the text, it could be helpful to refer to specific places in the Online Methods. In particular:

*line 103, what distance metric was used?

*line 108-110, states that the projection method (ie distance matrix clustered with MDS) gave higher resolution than the population clustering (ie presumably the hierBAPS mapped onto the UPGMA) but the specific result referred to here is not clear. Do the authors mean the 3 clusters in Suppl Fig1 vs the 2 larger clusters in Suppl Fig2? What is the consequence of 2 vs 3 clusters on their analysis?

*line 110-112, these pre-processing steps need some rationale. Very rare (<1%) or very common (>99%) kmers are excluded, which is fine. However, what is being done with the Chi squared test - are only the significant kmers in Chi-squared retained and how is the threshold justified? How does this change the P-value distribution (eg in a P-P plot)? The authors need to clarify this issue and show the P-value distribution before and after this pre-processing. Also in several places (eg lines 170,335), the authors refer to "informative" kmers; presumably this means the pre-processed kmers, but this term was not defined.

*line 428-431, are the authors suggesting that a single P-value cut-off ($10E-8$) be used for all samples, or is this threshold customized for different samples where the data could differ considerably (ie larger or smaller genomes with more or less complexity, giving different distributions of kmers)?

2) The application of the method to the *Streptococcus pyogenes* data was not convincing. My view is that this aspect of the paper would be greatly strengthened if the instructions on lines 318-320 were done here. For example, the authors could use another large sample of *S. pyogenes* genomes and show these two obscure loci replicate for invasiveness. This could be addressed in a meta-analysis context, as the authors make a point to facilitate meta-analysis (lines 142-146) *line 293, again, in the text the authors should indicate the nature of the 675 genome sample, and refer to the Online Methods. However, even in the Methods there is no reference cited for these genomes.

*After correcting for geographic location, the Tn916 hit disappears and two other loci appear. Is the pepF gene actually a hit, or is it only the adjacent genes? Relating to the other hit, nrdI and IgG binding protein H, the references cited are quite old -are there more recent results supporting their role in virulence?

Other comments:

3) line 60, other kmer approaches have been published but were not considered by the authors: kSNP by Gardner and Hall (2013 Plos One 8:e81760).

4) lines 194-195, should probably explain that the numbers in parentheses are the number of kmers, and indicate how far out the significant kmers can reach (eg 1 Mb away?)

5) lines 212-222, the erythromycin resistance results. First, the erm and mef resistances do not perform the same function - one is a methylase the other is an efflux pump. Secondly, what relevance are the permease, prfC, gatA, and ICE hits shown in Table1 - all false-positives?

6) reference sequence related:

*line 265 muddled wording, delete "SNPs have been called against"

*the alternative laid out on lines 268-273 is too speculative, delete

*line 275, the issues with respect to reference sequences are not "avoided" with this approach if you want to learn what the kmers represent. However, they may be streamlined somewhat.

* one issue I did not see addressed are kmers that map equally well in different places of the genome

7) line 434, what went wrong in the genome simulations to give an overly diverse genome sample that is 80% shorter than a pneumococcal genome? Were only invariant sites simulated?

Reviewer #2 (Remarks to the Author):

The paper describes a method for GWAS on microbial genomes based on k-mers. The advantage of the method as compared to previous methods is the scalability and the independence of genome alignments. The method is implemented in a stand-alone application (which I have not tested), which is a big advantage. Using linear models seems like the right choice and I find that the simulation results prove that it is a good strategy. One of the crucial points of the paper is the control for population structure, which I find should be better explained - see below. The paper is generally very well written.

For me this is a new subject and for me and many other readers, I think the paper would improve with a slightly longer introduction to the type of GWAS addressed (para 2 of intro).

Multidimensional scaling:

Please try to explain why the MDS controls for population structure, motivate this approach and give the reader some intuition for why this is smart. If it is a standard method, give some references. Help the reader by telling us that the distance matrix is an estimated number of non-shared k-mers. Explain why you use this particular distance, which seems to have an unfortunate scaling with genome sizes (why not normalize by tot. number of k-mers?). In the MDS you say that you project into 3 dimensions (why 3?) - then that the number of dimensions can be specified (starting line 385) - please make clear. The abbreviation MDS is undefined - please define it first time multi-dim. scaling is mentioned.

Although details are available in the methods section, it would be good to add a few details in this paragraph (how are the limits set):

Before testing for association we filter k-mers based on their frequency and unadjusted p-value to reduce false positives from testing underpowered k-mers and reduce computational time.

In methods, please specify what you do if the p-value of the chi-square/Welch test doesn't meet the cut-off (remove k-mer, I assume).

We wish to thank the reviewers for their helpful and constructive comments, and also their enthusiasm for the software package we have released. The comments have definitely helped to create an improved version of the paper which we attach here as a revision.

We would also like to note that as the manuscript has been transferred between journals some formatting changes are required in this second version (most notably, a single methods section after the discussion). This reformatting has helped address some of the concerns about clarity of the method, and we note where this is the case in our responses below. We have also added three authors who collected and extracted the DNA from the *S. pyogenes* strains we sequenced and analysed, who were omitted in the first version.

In addition to the changes suggested by reviewers we have also improved the usability of the code and documentation, based on communications with users we have been involved in since releasing the first version of the software when we first submitted the manuscript. This has included the development of a self-contained virtual machine with no dependencies; a statically compiled version of the software; the addition of scripts to help with interpreting significant k-mers; an improved wiki with usage and installation instructions; the addition of unit tests; adding more information on covariates into the output of SEER; more robust checks for difficult to invert matrices; and invoking Firth regression in a wider range of cases where normal logistic regression is underpowered. Collectively, we feel that this has enhanced the applicability of this novel toolset to the wider scientific community.

Reviewer #1 (Remarks to the Author):

This paper appears to offer several improvements on the general kmer approach to GWAS with large samples of bacterial genomes. Simulations are used to characterize the method, and two streptococcal samples are used to demonstrate its use with real data. My major concerns should be addressable with moderate effort. Overall, I feel the approach could be a useful addition to the bacterial GWAS toolbox.

Major concerns:

1) The authors did not clearly communicate some of the details related to the population structure correction, pre-processing of kmers, and P-value threshold for significance. During the presentation in the text, it could be helpful to refer to specific places in the Online Methods. In particular:

We address the specific concerns individually below. As we now have a single methods section the presentation should be clearer, and we have been careful to reference methods when they first appear in the results.

**line 103, what distance metric was used?*

This line now immediately proceeds the definition of the distance matrix (at line 428).

**line 108-110, states that the projection method (ie distance matrix clustered with MDS) gave higher resolution than the population clustering (ie presumably the hierBAPS mapped onto the UPGMA) but the specific result referred to here is not clear. Do the authors mean the 3 clusters in Suppl Fig1 vs the 2 larger clusters in Suppl Fig2? What is the consequence of 2 vs 3 clusters on their analysis?*

We agree that we communicated this point unclearly in the first version of the manuscript, as both reviewers have mentioned. We believe we have improved this section by writing it all as one section in the methods, adding supplementary figure 13, and rewording (lines 425-455).

Our first result, guided by previous work presented in reference 36 ('Random projection based clustering for population genomics'), is that the distances calculated by non-shared k-mers can achieve the same clustering as BAPS applied to a core gene alignment (which has been used in previous GWAS methods).

We then project this matrix into a lower number of dimensions, with the position of each sample in this space used as covariates in the regression. We discuss the effect of the choice of number of dimensions on sensitivity and specificity, and show in supplementary figures 13 and 14 (13 is added in the revision) how this may be chosen for the datasets covered in this manuscript.

**line 110-112, these pre-processing steps need some rationale. Very rare (<1%) or very common (>99%) kmers are excluded, which is fine. However, what is being done with the Chi squared test - are only the significant kmers in Chi-squared retained and how is the threshold justified? How does this change the P-value distribution (eg in a P-P plot)? The authors need to clarify this issue and show the P-value distribution before and after this pre-processing.*

We have added three relevant supplementary figures (8-10) two of which show P-P plots as suggested, as well as the marginal p-value distributions for the χ^2 test and (adjusted) logistic regression. These plots are discussed in the methods (lines 409-423), where we also discuss the justification for the χ^2 cut-off. Briefly, we find that k-mers below this cut-off do not map to the gene causal for the phenotype in our simulated data, and additionally present evidence which suggests their association is mainly lineage driven.

Also in several places (eg lines 170,335), the authors refer to "informative" kmers; presumably this means the pre-processed kmers, but this term was not defined.

In the original submission this was defined on lines 352-356. We have now included a reference to this definition where it appears in the results (line 158), and removed the jargon from the discussion for clarity.

**line 428-431, are the authors suggesting that a single P-value cut-off (10E-8) be used for all samples, or is this threshold customized for different samples where the data could differ considerably (ie larger or smaller genomes with more or less complexity, giving different*

distributions of kmers)?

This threshold should indeed be customised based on the sample set/genome in question. We believe the most robust way to do this is through permutation testing, as we performed here. This is, however, computationally intensive and potentially impractical for very large datasets. The result we present showing that the threshold given by permutation testing is very close to the threshold derived from considering all sites as variable is therefore suggested as a practical alternative for users. If variation within the dataset, or of a similar dataset, has been characterised before this may also guide the number of tests to adjust for.

We have laid out this reasoning in the methods to address this point (lines 512-531).

2) The application of the method to the Streptococcus pyogenes data was not convincing. My view is that this aspect of the paper would be greatly strengthened if the instructions on lines 318-320 were done here. For example, the authors could use another large sample of S. pyogenes genomes and show these two obscure loci replicate for invasiveness. This could be addressed in a meta-analysis context, as the authors make a point to facilitate meta-analysis (lines 142-146)

This report actually used two unrelated *S. pyogenes* datasets, one published dataset from Kenya and one newly sequenced dataset reported for the first time in this manuscript. The value of these unique *S. pyogenes* datasets are that they are population studies which are not confounded by prior knowledge of strain sequence type. We have reviewed five collections of *S. pyogenes* genomes that have been published on, but they all analyse only a single sequence type, and either have no metadata on invasiveness available or are very small sample collections. Despite direct requests, we have been unsuccessful in obtaining the relevant metadata on these published datasets to conduct further analyses. Therefore, while we can see that some form of validation might be necessary were this the primary result of the manuscript, there are no such data currently available and assembling a suitable new tertiary isolate collection would require extensive clinical resources and time, most likely several years. Since we have already included a detailed analysis of the causal variant discovery for the antibiotic resistance phenotypes and for synthetic data, we feel that validation of the identified *S. pyogenes* loci can reasonably be left for future research.

**line 293, again, in the text the authors should indicate the nature of the 675 genome sample, and refer to the Online Methods. However, even in the Methods there is no reference cited for these genomes.*

The appropriate references have been added, and the genomes are discussed when they first appear in the results rather than in the methods (line 293).

**After correcting for geographic location, the Tn916 hit disappears and two other loci appear. Is the pepF gene actually a hit, or is it only the adjacent genes?*

pepF is the hit, the legend labelling the figure was originally misleading. We have now corrected the text of the legend of supplementary figure 7, and relabelled supplementary figures 6 and 7.

Relating to the other hit, nrdI and IgG binding protein H, the references cited are quite old -are there more recent results supporting their role in virulence?

We have performed a more thorough literature search, and have added three more up-to-date references on the role of IgG binding protein H (*sph*) expression in *S. pyogenes* virulence. These references report upregulation of *sph* during invasive disease in a mouse model, and no expression during culture conditions. We believe our findings support these preliminary observations and envisage renewed interest in this area of *S. pyogenes* pathogenesis.

Other comments:

3) line 60, other kmer approaches have been published but were not considered by the authors: kSNP by Gardner and Hall (2013 Plos One 8:e81760).

We have added this reference (line 88), and some others, which discuss other uses of k-mer approaches in bacterial genomics. We hope this improves the introduction to the paper, along with our addition suggested by reviewer 2.

4) lines 194-195, should probably explain that the numbers in parentheses are the number of kmers, and indicate how far out the significant kmers can reach (eg 1 Mb away?)

We have added this explanation, and have indicated the overall coverage of the significant k-mers with the new supplementary figure 2. There are hits all over the genome, but may be expected for some traits as there is genome wide LD (see Fig 1 panel e in Chen and Shapiro and Fig S4 in Chewapreecha *et. al.* 2014).

5) lines 212-222, the erythromycin resistance results. First, the erm and mef resistances do not perform the same function - one is a methylase the other is an efflux pump.

Thanks for pointing this out, we have corrected the text (line 202).

Secondly, what relevance are the permease, prfC, gatA, and ICE hits shown in Table1 - all false-positives?

We have added a paragraph discussing these hits (lines 210-218). We believe that the ICE is in LD with the causal genes mentioned above, *prfC* and *gatA* may be compensatory mutations and the permease may be in LD with resistance mutations in *folP*. The latter two interpretations are speculation, and the hits may just be false-positives as suggested. We make this clear in the text.

6) reference sequence related:

**line 265 muddled wording, delete "SNPs have been called against"*

**the alternative laid out on lines 268-273 is too speculative, delete*

**line 275, the issues with respect to reference sequences are not "avoided" with this approach if you want to learn what the kmers represent. However, they may be streamlined somewhat.*

Thanks for these points, these edits have been included.

** one issue I did not see addressed are kmers that map equally well in different places of the genome*

We now address this issue in the methods (lines 561-566), with relation to where it was seen in our dataset.

7) line 434, what went wrong in the genome simulations to give an overly diverse genome sample that is 80% shorter than a pneumococcal genome? Were only invariant sites simulated?

We used a gamma + invariant sites model as the distribution of rate heterogeneity among sites. As we didn't have estimates for the parameters of this distribution directly from our data, we used the estimate given by ALF. The resulting gamma distribution must have a longer tail than the real data, as some sites vary at high frequency. This creates many low-frequency k-mers.

As the simulation is computationally expensive to run, we reasoned that rather than running it lots of times with different parameters until a k-mer distribution identical to the observed data was reached we could use the original result as these low frequency k-mers would be filtered out in the common variation associations we are testing. 24.7M k-mers pass frequency filtering from the real data, whereas 12.7M pass from the simulated data – while this isn't quite the linear scaling expected with genome length (which would predict around 7M k-mers) the amount of common variation at the gene level is similar to real data.

For the purposes we use the simulations for, a gene driven association at different ORs, we believe this result is still an appropriate test. The genomes are related by a real phylogenetic tree with convergent evolution, gene loss and horizontal gene transfer – which are the key features being tested.

We have added this discussion verbatim to the manuscript in supplementary methods (as it is quite technical, and not central to the rest of the results).

Reviewer #2 (Remarks to the Author):

The paper describes a method for GWAS on microbial genomes based on k-mers. The advantage of the method as compared to previous methods is the scalability and the independence of genome alignments. The method is implemented in a stand-alone application (which I have not tested), which is a big advantage. Using linear models seems like the right choice and I find that the simulation results prove that it is a good strategy. One of the crucial points of the paper is the control for population structure, which I find should be better explained - see below. The

paper is generally very well written.

For me this is a new subject and for me and many other readers, I think the paper would improve with a slightly longer introduction to the type of GWAS addressed (para 2 of intro).

We have written an extra couple of paragraphs in the introduction which explain the background and problems faced by bacterial GWAS in more detail, including relevant up-to-date references (lines 53-70).

Multidimensional scaling:

Please try to explain why the MDS controls for population structure, motivate this approach and give the reader some intuition for why this is smart. If it is a standard method, give some references.

This method is analogous to the use of principal components as covariates to correct for ancestry in human genetics GWAS studies, but has the advantage that no core gene alignment or SNP calling is needed (i.e. it can be applied directly to the output of the k-mer counting program). We have added in description and references to appropriate papers (line 429) which show this method is appropriate in the context of human ancestry to motivate its use to control for bacterial ancestry and relatedness. Reference 36 also supports our use of this approach.

Help the reader by telling us that the distance matrix is an estimated number of non-shared k-mers.

This is a clear interpretation! Added at line 441.

Explain why you use this particular distance, which seems to have an unfortunate scaling with genome sizes (why not normalize by tot. number of k-mers?).

This suggestion is a very good one, as it has a more natural interpretation should users wish to apply their own transform to the distance matrix, and doesn't require scaling at a later stage. The software therefore now applies the suggested normalisation (commit 58879202cfbba04b67f6c55f2d2a9e9bf348a0b0). However, this will not have directly affected the results presented in the paper, as the MDS values were normalised at a later stage to be on the same scale as the k-mer frequency vector (see kmDestruct.cpp:53-66 @ commit de956943b4b06a1c2a9b0f72989e322411e32308).

In the MDS you say that you project into 3 dimensions (why 3?) - then that the number of dimensions can be specified (starting line 385) - please make clear.

We have clarified this, and in addition improved this section in line with reviewer 1's first comment. This also addresses why we chose to use three dimensions.

The abbreviation MDS is undefined - please define it first time multi-dim. scaling is mentioned.

Added at line 428.

*Although details are available in the methods section, it would be good to add a few details in this paragraph (how are the limits set):
Before testing for association we filter k-mers based on their frequency and unadjusted p-value to reduce false positives from testing underpowered k-mers and reduce computational time.*

This issue is addressed in our response to reviewer 1's first comment (lines 409-423), and in the overall reformatting of methods and results.

In methods, please specify what you do if the p-value of the chi-square/Welch test doesn't meet the cut-off (remove k-mer, I assume).

Added at line 406.

REVIEWERS' COMMENTS:

Reviewer #1 (Remarks to the Author):

This revised paper outlines a novel method for bacterial GWAS that I believe to be a valuable contribution to the field. The revisions have clarified important details of their method, and some details of the simulations that were used to evaluate their method. The analysis of the *Strep pyogenes* data is also improved as the source of the data is now clear and their results are placed in context. The response to prior review comments was very thorough. Only two minor (negligible) edits are suggested:

1. Add indels to the list of polymorphisms that can be associated with phenotype on lines 84-85, as this type of variation can also be probed by this method (line 128).
2. Reference 2 and 9 are duplicated.

Reviewer #2 (Remarks to the Author):

The authors have addressed all the points I raised in the first review, and I have no further remarks.

Reviewer #1 (Remarks to the Author):

*This revised paper outlines a novel method for bacterial GWAS that I believe to be a valuable contribution to the field. The revisions have clarified important details of their method, and some details of the simulations that were used to evaluate their method. The analysis of the *Strep pyogenes* data is also improved as the source of the data is now clear and their results are placed in context. The response to prior review comments was very thorough.*

Only two minor (negligible) edits are suggested:

1. Add indels to the list of polymorphisms that can be associated with phenotype on lines 84-85, as this type of variation can also be probed by this method (line 128).

Added in revision.

2. Reference 2 and 9 are duplicated.

Fixed in revision.

Reviewer #2 (Remarks to the Author):

The authors have addressed all the points I raised in the first review, and I have no further remarks.